# A Multi-Rate Simulation Strategy Based on the Modified Time-Domain Simulation Method and Multi-Area Data Exchange Method of Power Systems

Ruotian Yao [1,†], Qi Chen [2,3,†], Hao Bai [1], Chengxi Liu [2,3,*], Tong Liu [1], Yongjian Luo [2,3] and Weichen Yang [1]

1   Electric Power Research Institute of CSG, Guangzhou 510663, China; yaort@csg.cn (R.Y.); baihao@csg.cn (H.B.); liutong@csg.cn (T.L.); yangwc@csg.cn (W.Y.)
2   School of Electrical Engineering and Automation, Wuhan University, Wuhan 430072, China; chenqicq@whu.edu.cn (Q.C.); yongjianluo@whu.edu.cn (Y.L.)
3   Hubei Engineering and Technology Research Center for AC/DC Intelligent Distribution Network, Wuhan 430072, China
*   Correspondence: liuchengxi@whu.edu.cn
†   These authors contribute equally to this work.

**Abstract:** Accurate modeling for power-electronic devices requires power systems to be simulated with considerably small step sizes (typically several microseconds), which causes unnecessary computational burden and reduces efficiency, especially for large-scale power systems. To achieve a balance between simulation precision and efficiency, this paper introduces an innovative multi-rate interface strategy based on the modified time-domain simulation (TDS) method and multi-area data exchange method. The modified TDS method transforms the initialization process into exchange of electric data among different subsystems, while the multi-area data exchange method is able to ensure numerical stability and simulation universality during the multi-rate simulation. The proposed strategy provides a robust interface that allows different subsystems to be engaged in simulations with different step sizes while exchanging data. To validate this strategy, simulations on an integrated System of IEEE 14-bus and 33-bus systems is conducted. In addition, the strategy is further applied to a real-world scenario of the Subsystem in the Guangxi Power Grid in China. Analysis of the results indicates that the proposed multi-rate fast simulation strategy can significantly boost simulation efficiency while maintaining accuracy, which marks a notable improvement compared with the traditional single step size simulation.

**Keywords:** time-domain simulation; multi-area data exchange; multi-rate interface; time synchronization; interpolation correction

## 1. Introduction

With the high penetration of distributed energy resources and power electronics devices, modern power systems feature dynamic characteristics with multiple time scales. Due to the need for precise control of power electronic devices in microsecond timescales, a considerably small step size is normally required for simulating power electronic systems [1–3]. Because of the presence of equipment, such as synchronous generators and power electronics devices, the components in the grids have different time constants, demonstrating both fast and slow dynamic characteristics. The simulation step size of the former is typically in the range of microseconds or even nanoseconds, which may lead to inefficiency, especially for large-scale power systems. The simulation of the latter, on the other hand, generally requires a step size of tens of milliseconds, which is faster but cannot accurately model the dynamic characteristics of power electronic devices. Therefore, one of the solutions for the simulation of modern power systems is to adopt a multi-rate interface strategy [4], which uses different simulation step sizes for network areas with

different time constants, and for different areas being connected through appropriate data exchange [5,6]. By adopting the above strategy, accurate simulation results can be obtained while expediting the simulation process for large-scale power systems.

However, due to the inherent limitations of commonly used multi-rate simulation algorithms, it is difficult to simultaneously achieve both simulation accuracy and efficiency [6]. The original multi-rate method, proposed by Gear and Wells, is designed to solve systems of ordinary differential equations [7]. This method uniquely combines different simulation time steps and integration methods for various variables, coupling fast and slow variables through techniques such as interpolation. In modern power systems, there are two different dynamic response speeds suitable for the multi-rate method. However, the multi-rate algorithm proposed by [7] cannot be applied directly because of the strong voltage and current constraints between different subsystems [8]. The fast-priority and slow-priority algorithms use extrapolation to synchronize different time steps [9], but this compromises the simulation's accuracy and stability. The multi-rate simulation using the slack variable method strictly enforces the voltage and current constraints through iteration [10,11], but its simulation efficiency is limited. The multi-port Thevenin equivalent algorithm based on full implicit integration also improves simulation efficiency [11]. The above methods are all multi-rate methods based on variables derived from the original method proposed by [7].

With the large-scale integration of power electronic devices, multi-rate algorithms have developed to the System level [12,13]. In [14], a multi-rate parallel transient simulation technique for large-scale distribution networks is proposed, based on network decomposition. This method employs interface equivalent models and adaptive variable step size techniques to accelerate parallel simulations. To reduce errors and ensure accuracy, it also adopts interpolation methods and employs a truncation error control strategy with the adaptively adjusted step size. In [15], the authors introduce a novel multi-rate hybrid solver for AC/DC systems to address the nonlinear issues. The solver applies both iterative and non-iterative solvers with different time steps to the decomposed subsystems and has been successfully validated on a real-time simulation platform. Reference [16] presents a multi-rate method for simulating multi-physical systems with a wide range of time scales in the context of an Electric Storage Unit (ESU) for all-electric ferries. This method is capable of handling the high interdependence between fast and slow state variables, thereby enhancing simulation efficiency. The authors of [17] integrate a multi-scale algorithm for the faster electromagnetic and slower electromechanical dynamic simulations based on matrix exponential functions and conduct numerical studies, including simulations of large-scale wind farms. These above methods are generally conducted with MATLAB, which leads to comparatively lower versatility on other platforms. Therefore, a strategy built on the upper layer is needed to improve the versatility and adaptability, and thus can be adapted to other software.

Currently, a wide range of commercially mature simulation and analysis software for power systems is available [18,19]. However, most of these solutions do not permit user modifications to the already built-in algorithms, posing significant challenges for further development. When information exchange between inter-systems is required, the above-mentioned constraint encounters the inability to alter the built-in integration algorithms used for computing time-domain responses. Moreover, there is a lack of mature multi-rate simulation programs. This gap hinders the users from choosing supported single-rate integration algorithms and adjusting the integration parameters to optimize simulation speed while ensuring accuracy. Consequently, a multi-rate interface strategy based on a modifiable multi-area data exchange method should be considered. With the management of initialization of dynamic simulations and setting up of strategy-based communication interfaces at higher architectural levels, this method alleviates the need for extensive knowledge or alterations of the underlying code, and enables efficient data interaction while retaining the advantages of the original simulation software. HELICS (Hierarchical Engine for Large-scale Infrastructure Co-Simulation), exemplary in the realm of co-simulation frameworks, facilitates concurrent simulation of various systems, including integrated

energy and physical systems. Each System is modeled independently yet interacts during the simulation process [20,21]. Similarly, FNCS (Framework for Network Co-Simulation) offers a foundational platform for the co-simulation of power systems and communication networks [22]. It allows disparate simulators to function within a unified environment, orchestrating their interactions and synchronization. However, the over-generalization and heavy reliance on a framework-centric approach in these systems cause a notable disconnect between the algorithms and practical applications, particularly evident in FNCS. This feature poses challenges for developers aiming to implement modifications for use in multi-rate simulations within these frameworks.

In view of the areas to be improved in the above research, this paper proposes a multi-rate interface technology based on a modified time-domain simulation method and a multi-area data exchange method. The main contributions are outlined as follows:

(1) To ensure the accuracy of multi-rate simulation, this paper proposes a modified TDS method. The proposed method modifies the built-in algorithm of the traditional TDS method, which is not applicable for multi-rate simulation. Without any approximation, this modification enables initial data of subsequent simulation to be obtained from information exchange between different subsystems after the first process. Therefore, high accuracy can be achieved in completing multi-rate simulations.

(2) To verify the versatility of the multi-rate interface, this paper proposes a multi-area data exchange method. It allows the interface data to be transferred between different subsystems, while maintaining high applicability and superiority of the original dynamic simulation algorithm. Furthermore, this method adopts interpolation technique to solve the numerical stability issues caused by varying tolerance levels of different systems for different exchanged data.

## 2. Multi-Rate Interface Strategy

This section introduces the multi-rate dynamic simulation for power systems. Firstly, the limitations of the traditional TDS method for multi-rate simulation are discussed. Then, a modified TDS method is proposed to apply on the multi-rate interfaces. The ability of this method to enhance simulation efficiency has already been theoretically verified. Moreover, approaches are presented to address numerical instability issues due to distinct tolerance thresholds for limits of varying systems. Finally, this paper introduces a multi-area data exchange method, which enables data exchange without changing the main dynamic simulation algorithm, making it more universally applicable for simulation software.

### 2.1. The Modified TDS Method

A multi-rate simulation was proposed, which involves dynamic simulations of systems on varying time scales, including rapid electromagnetic transients and slow mechanical transients [23]. However, traditional simulation software for dynamic simulation often underperforms due to the design of its inherent algorithm in handling multi-rate simulations of power systems, which might be attributed to the fact that their design and implementation mainly focus on steady-state and transient analysis of power systems, without specific optimization for multi-rate simulations. Such a limitation could impact the accuracy and efficiency of simulations involving dynamics across various time scales [24]. A thorough investigation into Python open-source software reveals that the initialization process for TDS fundamentally relies on the outcomes of power flow computations. This inherent design presents a significant obstacle when adapting it for multi-rate simulations. Specifically, if an attempt is made to initialize information interactively across different systems to facilitate multi-rate simulation at each step, the interface data input into the System would be superseded by the initialization data generated from each power flow result [25]. This poses a fundamental constraint, making the direct application impractical for multi-rate simulation scenarios.

In order to overcome the aforementioned problem, this paper presents a modified TDS method that enables multi-rate simulation. The modification entails a significant change

to the dynamic simulation program, which overcomes the inherent limitations in the TDS of the software's built-in algorithms. The initial exchange of the multi-rate simulation proceeds by running programs separately and interacting via input–output files. However, starting from the first exchange, Subsystem simulations do not initialize, and necessary data such as voltage are exchanged at the interface, with all other data remaining unchanged from the previous step size simulation. This ensures that the second exchange is based on the first, with subsequent processes following in an iterative fashion to complete the simulation. This change allows for the exchange of information between different systems, thus facilitating multi-rate simulation. This modification not only resolves the existing limitations without altering the original functionality of the software, but also expands its capabilities, transforming it into a more versatile tool for power System simulation. Below is the mathematical formulation of the modified TDS method for multi-rate simulation.

Suppose we have two subsystems, Subsystem 1 and Subsystem 2. Each Subsystem can be mathematically represented by a set of differential algebraic equations (DAEs) [26]:

For Subsystem 1:

$$\frac{dx_1}{dt} = f_1(x_1, y_1, u_1) \tag{1}$$

$$0 = g_1(x_1, y_1, u_1) \tag{2}$$

For Subsystem 2:

$$\frac{dx_2}{dt} = f_2(x_2, y_2, u_2) \tag{3}$$

$$0 = g_2(x_2, y_2, u_2) \tag{4}$$

where $x$ denotes the state variables, $y$ represents the algebraic variables, $u$ is the System input, and $f$, $g$ are System functions describing the dynamic and algebraic behaviors of the system, respectively.

In traditional dynamic simulation methods for power systems, System initialization is typically achieved by solving a power flow problem. The results of the power flow provide a steady-state solution that matches actual operating conditions, which will serve as the initial conditions for the dynamic simulation. This includes the initial voltage and phase angle for each bus, as well as the initial state of the generators, as shown in Equations (5) and (6):

$$x_1(0) = x_{1,\text{init}}, \ y_1(0) = y_{1,\text{init}} \tag{5}$$

$$x_2(0) = x_{2,\text{init}}, \ y_2(0) = y_{2,\text{init}} \tag{6}$$

In Equations (5) and (6), $x(0)$, $y(0)$ and $x_{\text{int}}$, $y_{\text{int}}$ represent the initial states of subsystems 1 and 2 for each simulation step and the state values obtained from the initialization process of the flow results, respectively. The numerical solution method used in this paper employs a simultaneous solution approach. Its basic process involves first using implicit integration formulas to algebraize the set of differential equations, which together with the set of algebraic equations form a simultaneous nonlinear equation set. Then, Newton's method is used to solve this set of nonlinear equations, thereby eliminating the need for repeated alternate solving of differential and algebraic equations, as expressed in Equations (7) and (8):

$$\begin{aligned} 0 &= \hat{q}_1(x_1(t + \Delta t), y_1(t + \Delta t), f_1(t)) \\ 0 &= g_1(x_1(t + \Delta t), y_1(t + \Delta t)) \end{aligned} \tag{7}$$

$$\begin{aligned} 0 &= \hat{q}_2(x_2(t + \Delta t), y_2(t + \Delta t), f_2(t)) \\ 0 &= g_2(x_2(t + \Delta t), y_2(t + \Delta t)) \end{aligned} \tag{8}$$

where $\hat{q}$ is a function dependent on the implicit numerical method used. Equations (7) and (8) are nonlinear and their solution is achieved using Newton's method. This involves iteratively calculating the increments $\Delta x^{(i)}$ and $\Delta y^{(i)}$ for the state and algebraic variables,

and then updating the actual variables [24]. During a given iteration denoted by $i$, the quantities $\Delta x^{(i)}$ and $\Delta y^{(i)}$ can be formulated by Equation (9) as:

$$\begin{bmatrix} \Delta x^{(i)} \\ \Delta y^{(i)} \end{bmatrix} = -\left[A_c^{(i)}\right]^{-1}\begin{bmatrix} \hat{q}^{(i)} \\ g^{(i)} \end{bmatrix}$$

$$\begin{bmatrix} x^{(i+1)}(t+\Delta t) \\ y^{(i+1)}(t+\Delta t) \end{bmatrix} = \begin{bmatrix} x^{(i)}(t+\Delta t) \\ y^{(i)}(t+\Delta t) \end{bmatrix} + \begin{bmatrix} \Delta x^{(i)} \\ \Delta y^{(i)} \end{bmatrix} \tag{9}$$

where $A_c^{(i)}$ is a matrix that depends on the algebraic and state Jacobian matrices of the system. In the trapezoidal rule, $A_c^{(i)}$ and $\hat{q}^{(i)}$ can be obtained, as shown in Equation (10):

$$A_c^{(i)} = \begin{bmatrix} I_{n_x} - 0.5\Delta t f_x^{(i)} & -0.5\Delta t f_y^{(i)} \\ g_x^{(i)} & g_y^{(i)} \end{bmatrix}_z$$

$$\hat{q}^{(i)} = x^{(i)} - x(t) - 0.5\Delta t\left(f^{(i)} + f(t)\right) \tag{10}$$

where $I_{n_x}$ is the identity matrix of the same dimension of the dynamic order of the DAE System and all Jacobian matrices, and $f^{(i)}$ are computed at the current point $\left(x^{(i)}(t+\Delta t), y^{(i)}(t+\Delta t), t+\Delta t\right)$.

As described earlier, in the proposed modified method, starting from the second electrical data exchange, the two systems do not need to perform initialization calculation, and directly exchange interface data in the interface program. For example, Subsystem 1 removes the initialization process and instead calls the interface data from the first exchange of Subsystem 2. Other state variables and algebraic variables remain unchanged from the results of the first simulation. Then, Subsystem 1 continues to run the second simulation, as shown in Figure 1.

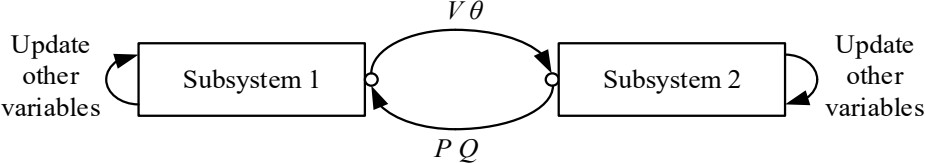

**Figure 1.** Description of data exchange between subsystems.

This ensures that the results of the second simulation are based on the first, without losing the stability and efficiency of the simulation. When Subsystem 2 transfers power to Subsystem 1, Equations (11) and (12) are utilized to update the electrical data from the previous exchange in advance, and then these are substituted into Equation (9) to uniformly update the state variables and algebraic variables of the system:

$$P_{1k}^{(i)}(t+\Delta t_1) = P_{2m}^{(i)}(t+\Delta t_2) \tag{11}$$

$$Q_{1k}^{(i)}(t+\Delta t_1) = Q_{2m}^{(i)}(t+\Delta t_2) \tag{12}$$

Similarly, when Subsystem 1 transfers voltage magnitude and phase angle to Subsystem 2, Equations (13) and (14) are used to pre-update Equation (9):

$$V_{2m}^{(i)}(t+\Delta t_2) = V_{1k}^{(i+1)}(t+\Delta t_1) \tag{13}$$

$$\theta_{2m}^{(i)}(t+\Delta t_2) = \theta_{1k}^{(i+1)}(t+\Delta t_1) \tag{14}$$

Assuming bus $k$ of Subsystem 1 is connected to bus $m$ of Subsystem 2, $P$, $Q$, $V$, and $\theta$ represent the active power, reactive power, voltage magnitude, and phase angle at the interface, respectively. $\Delta t_1$ and $\Delta t_2$ are the time steps for Subsystem 1 and Subsystem 2,

respectively. Typically, in the multi-rate simulation method applied to power System analysis, this simulation software offers the functionality to customize the simulation step size for each subsystem. Therefore, the step size for integrating the differential equations can be selected based on the dynamic characteristics of each subsystem, thereby enhancing the simulation efficiency.

### 2.2. Efficiency Improvement Analysis and Interpolation Correction

The critical consideration for data exchange of multi-rate interface is time synchronization [27]. However, the proposed method is markedly different from traditional multi-rate methods as it avoids the need for extrapolation approximation calculations. The essence of this approach is the flexible configuration of simulation times and time steps provided by the mature simulation software. In other words, the computational process of the proposed method is sequential, but it ensures the accuracy of the simulation. More specifically, once the simulation of Subsystem 1 is completed, it can be temporarily paused, allowing Subsystem 2 to finish its simulation at a finer time step. Thus, the information exchange can be conveniently carried out again. This method ensures that the data exchange mechanism in the simulation software operates along two independent paths, with information exchange occurring only at the coupling points between the two systems. Consequently, the challenge of time synchronization—a typically significant concern in multi-rate simulations—is effectively mitigated in this study.

It is noteworthy that, once System stability is ensured, the efficiency of the simulation can be enhanced with the reduction of the System matrix order. For instance, if a matrix of order $(2n)^2$ is divided into two subsystems with orders of $(0.5n)^2$ and $(1.5n)^2$, respectively, the overall efficiency, due to the serial process being entirely dependent on the $2.25n^2$ system, shows a remarkable improvement in simulation efficiency.

However, this introduces the issue of numerical instability caused by the different sensitivities of different systems to port values due to different step sizes [28]. In line with Section 2.1, consider two distinct systems, designated as Subsystem 1 and Subsystem 2, each operating at disparate temporal resolutions denoted by $\Delta t_1$ and $\Delta t_2$, such that $\Delta t_1 > \Delta t_2$ The data exchange between these systems is facilitated via electrical parameters including power ($P$), reactive power ($Q$), voltage ($V$), and phase angle ($\theta$). The state of System 1 at a given instant $t + \Delta t_1$ is described by $S_1(t + \Delta t_1)$, whereas the state of System 2 at $t + \Delta t_2$ is characterized by $S_1(t + \Delta t_2)$. Owing to this temporal granularity mismatch, there may be occasions when System 2 necessitates the state information from System 1 at intermediary points, specifically $t + k\Delta t_2$ where $k$ is a natural number and satisfies $k\Delta t_2 < \Delta t_1$. Employing the state values of $S_1(t)$ directly at these points might engender numerical instabilities, considering the values may not faithfully convey the actual state of System 1 at the instants $t + k\Delta t_2$.

In addressing the temporal discrepancy inherent to the interaction between System 1 and System 2, which operate under disparate time step resolutions $\Delta t_1$ and $\Delta t_2$ respectively, an interpolation method is necessitated. This approach seeks to estimate the state of System 1 at an intermediary temporal mark, denoted as $t + k\Delta t_2$, which lies between two consecutive discrete states $S_1(t)$ and $S_1(t + \Delta t_1)$. Designating $I$ as the interpolation operator, the estimated state of System 1 at any given point $t + k\Delta t_2$ is thus rendered by $I(S_1(t), S_1(t + \Delta t_1), k\Delta t_2)$. This interpolation operator can manifest in various formulations, such as linear, polynomial, or spline-based interpolation methodologies [29,30]. For instances necessitating linear interpolation, the operator $I$ can be mathematically articulated as Equation (15):

$$I(S_1(t), S_1(t + \Delta t_1), k\Delta t_2) = S_1(t) + \left(\frac{k\Delta t_2}{\Delta t_1}\right) \cdot (S_1(t + \Delta t_1) - S_1(t)) \tag{15}$$

With this construct, System 2 can employ $I(S_1(t), S_1(t + \Delta t_1), k\Delta t_2)$ to effectuate an update of its state at the precise time point $t + k\Delta t_2$, thereby attenuating the numerical instabilities promulgated by the incongruence of the time steps.

To analyze the stability of the interpolation method, the local truncation error (LTE) and global error (GE) for states $S_1$ and $S_2$ can be considered [31]. For Systems 1 and 2, the LTE can be expressed as Equation (16):

$$LTE_1 = S_1(t + \Delta t_1) - (S_1(t) + \Delta t_1 \cdot f_1(S_1(t), t))$$
$$LTE_2 = S_2(t + \Delta t_2) - (S_2(t) + \Delta t_2 \cdot f_2(S_2(t), t)) \tag{16}$$

where $f_1$ and $f_2$ are functions describing how the states of Subsystems 1 and 2 evolve over time. The global error, GE, can be computed as Equation (17):

$$GE_{1,i+1} = GE_{1,i} + LTE_{1,i}$$
$$GE_{2,j+1} = GE_{2,j} + LTE_{2,j} \tag{17}$$

where $i$ and $j$ represent the iterative steps for Subsystems 1 and 2, respectively. It is essential to ensure that interpolation does not introduce an increase in the GE, which can be ascertained by contrasting the GE observed with and without employing interpolation.

### 2.3. Multi-Area Data Exchange Method

Based on the modified TDS method, the multi-rate simulation focuses on the methods of exchanging input and output data. Suppose Subsystem 1 is a fast-dynamic system, and Subsystem 2 is a slow-dynamic system. The two subsystems are connected at a certain bus and run the TDS on it with different step sizes. This paper assumes that the data exchanged between the fast and slow dynamic systems include the amplitude and phase angle of the bus voltage, as well as the active and reactive power injected at the bus. Considering the connections between transmission and distribution networks, when Subsystem 1 is simulated independently, Subsystem 2 is treated as a PQ load, whereas when Subsystem 2 is left alone, Subsystem 1 is regarded as a Thevenin equivalent voltage source. The following are the main steps for data exchange:

(1)  The fast-dynamic System operates a dynamic simulation with a large step size of 10 ms. Assume a short-circuit fault is applied at 1.0 s and it lasts until 1.01 s, resulting in an output file. The amplitude and phase angle of the voltage are extracted from this output file and inputted as the injected power at the Slack bus in the slow dynamic system's input file. The slow dynamic System then runs a dynamic simulation with a small step size of 1 ms using this input file. This input file contains a minor disturbance to simulate the impact from Subsystem 1. Similarly, another output file is obtained, which provides the active and reactive power of the bus.

(2)  In general simulation software, the dynamic simulation program is called the TDS. The power-based modified TDS program for Subsystem 1 is designated as TDS1, and for Subsystem 2, it is designated as TDS2. The power from the bus in the previous step's output file is injected into Subsystem 1, while other variables remain unchanged from the previous step. On this basis, all variables are used in TDS1 for the second step of dynamic simulation. The output includes the amplitude and phase angle of the interface bus voltage, with interpolation processing and consideration of numerical stability issues during the exchange process.

(3)  Subsystem 2 receives the bus voltage transmitted by Subsystem 1 and then runs TDS2 for subsequent simulations.

(4)  By continuously repeating steps (2) and (3), a complete cycle of multi-rate simulation can be achieved.

So far, all aspects of multi-rate interface strategy based the modified TDS method and multi-area data exchange method have been clarified. Following are the detailed steps of the proposed strategy, as shown in Figure 2.

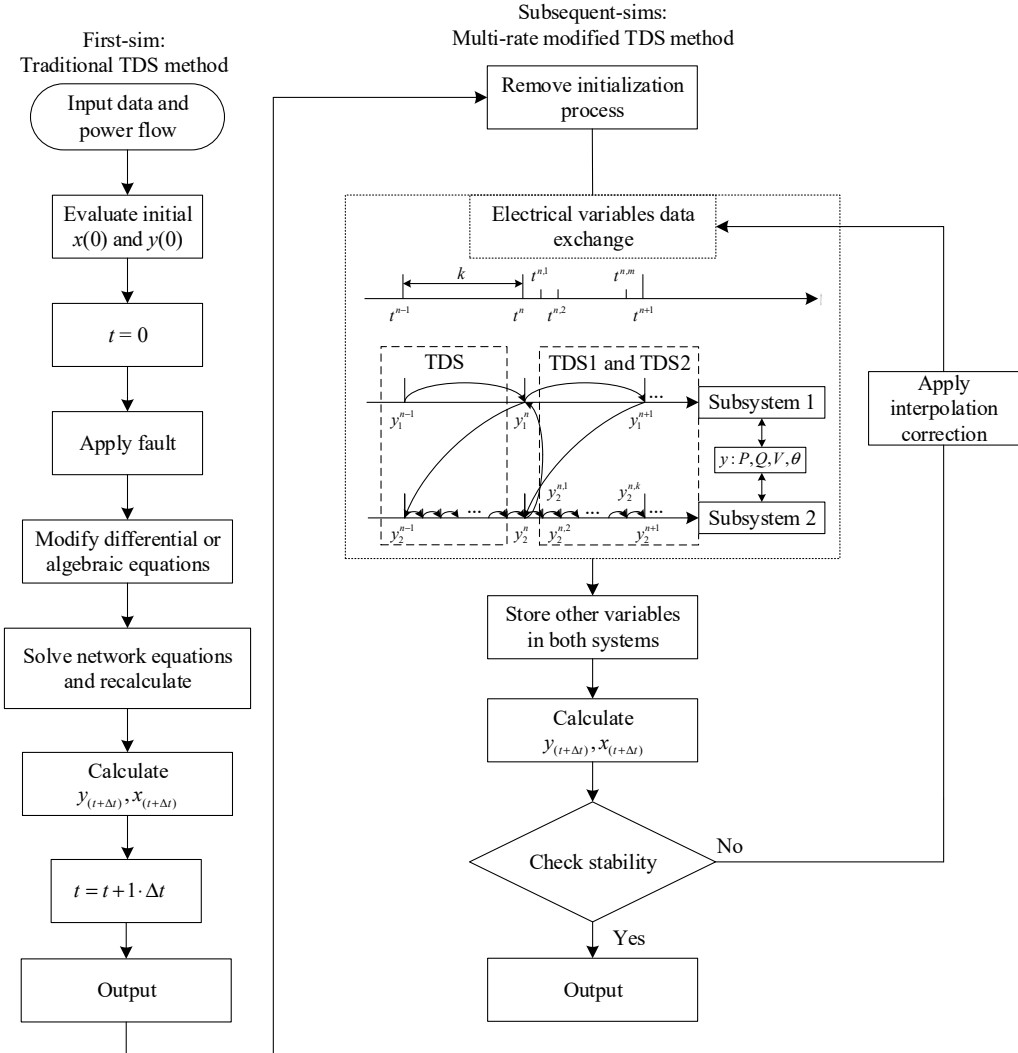

**Figure 2.** Detailed steps of the proposed strategy.

## 3. Case Studies

The integrated System of IEEE 14-bus and 33-bus systems is used for the case study. The simulation is performed in Python software 3.9 called WHU-Dis developed for a distribution network by Wuhan University on a personal computer with a 2.9 GHz i5 processor and 16 GB RAM. Firstly, an integrated simulation case will be constructed. Subsequently, the proposed interface strategy will be applied to perform multi-rate simulation. In the second case study of this paper, the interface is applied to a real-world scenario; specifically, the topology data of a certain power System in the Guangxi Power Grid in China, consisting of 212 bus, is utilized.

### 3.1. Case Study of the Integrated System of IEEE 14-Bus and 33-Bus Systems

3.1.1. Simulation Setup Description

The integrated simulation case involves connecting the starting bus of the IEEE 33-bus System to the 11th bus of the IEEE 14-bus System and its topology is shown in Figure 3. As discussed above, this method is intended to treat the IEEE 33-bus System as a load for the IEEE 14-bus system, and conversely, the IEEE 14-bus System acts as a Thevenin equivalent voltage source for the IEEE 33-bus system, thereby minimizing the influence of the simulation scenario differences between the separated and integrated simulations. Partial static parameters for the integrated System are available in Tables A1–A6 while additional static parameters can be found in references [32,33]. To validate the multi-rate

interface, an idealized setup is assumed where all generators of the IEEE 14-bus System are synchronous machines, and the IEEE 33-bus System is composed entirely of distributed photovoltaic (PV) generators. The synchronous machines are modeled using the classical model, and loads are represented by a constant power model. The dynamic parameters for the PV systems utilized in this study are listed in Tables A2 and A3. Simulation time steps are set at 10 ms and 1 ms, respectively, with a total simulation duration of 10 s. During the simulation, a three-phase short circuit fault is introduced at Bus 13, which is sustained for 0.1 s. All results are presented in per-unit (pu) values expect the frequency of PVs.

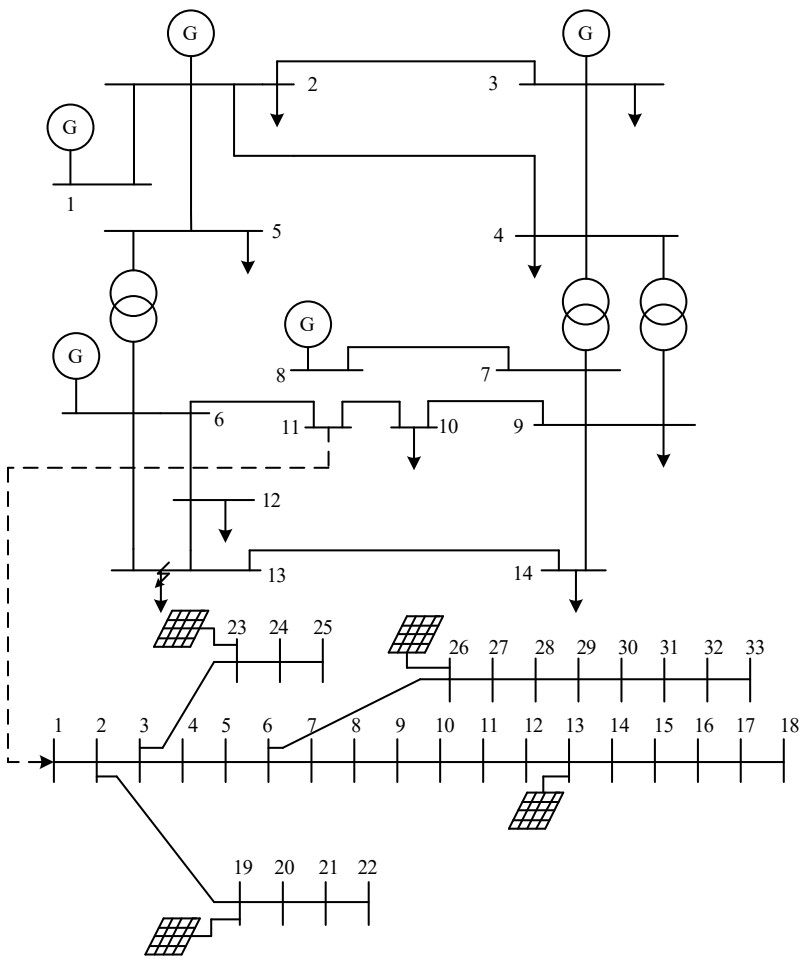

**Figure 3.** Topology of the integrated system.

### 3.1.2. Simulation Results Analysis

The large step size and the modified method were employed to calculate the output voltage and angle at the interface (Bus 11), as depicted in Figures 4 and 5. Correspondingly, Figure 6 shows the rotor speeds of the generator, while Figures 7 and 8 display the frequency and output power of the distributed PV, respectively, all determined using both simulation techniques. The simulation curves, extending from Figures 4–8 indicate that both approaches produced similar trends. However, the response with the large step size is noticeably rougher in dynamics, whereas the multi-rate method offers a more detailed depiction of the dynamic processes. Additionally, as seen from Table 1, the multi-rate method maintains the simulation efficiency. Thus, the proposed method achieves a balance between stability and efficiency in the simulation, situating itself between the two extremes.

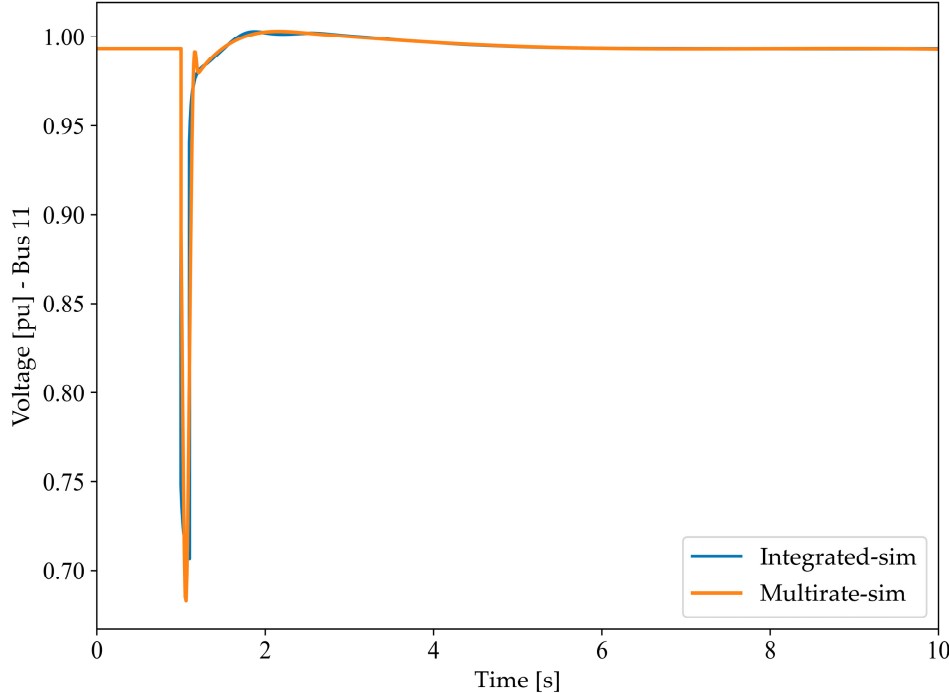

**Figure 4.** The voltage of interface (Bus 11) in the integrated and multi-rate simulations.

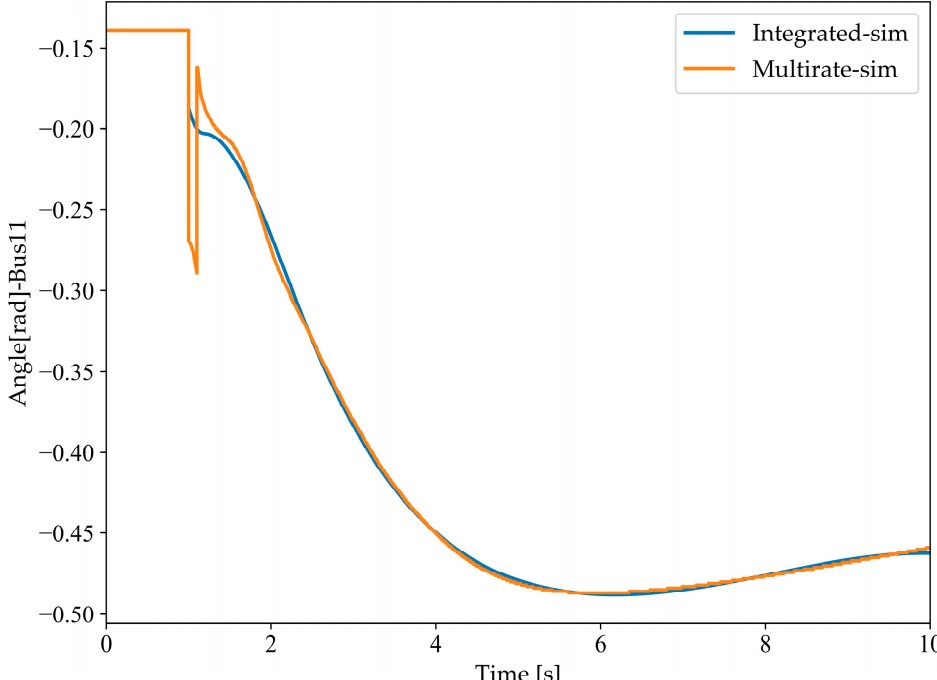

**Figure 5.** The angle of interface (Bus 11) in the integrated and multi-rate simulations.

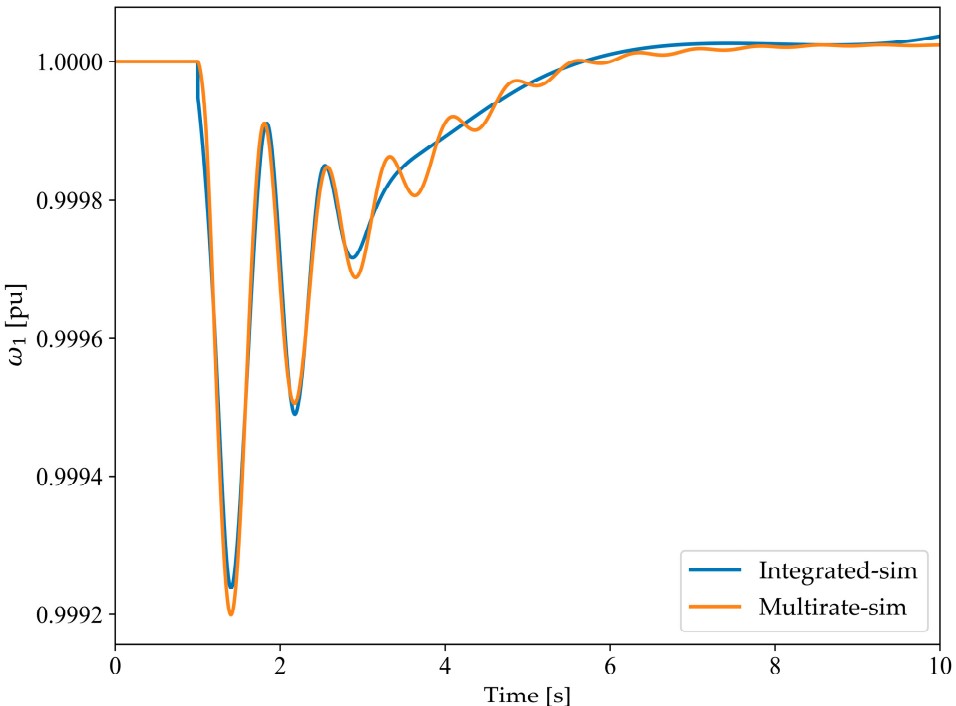

**Figure 6.** The rotor speed of slack bus in the integrated and multi-rate simulations.

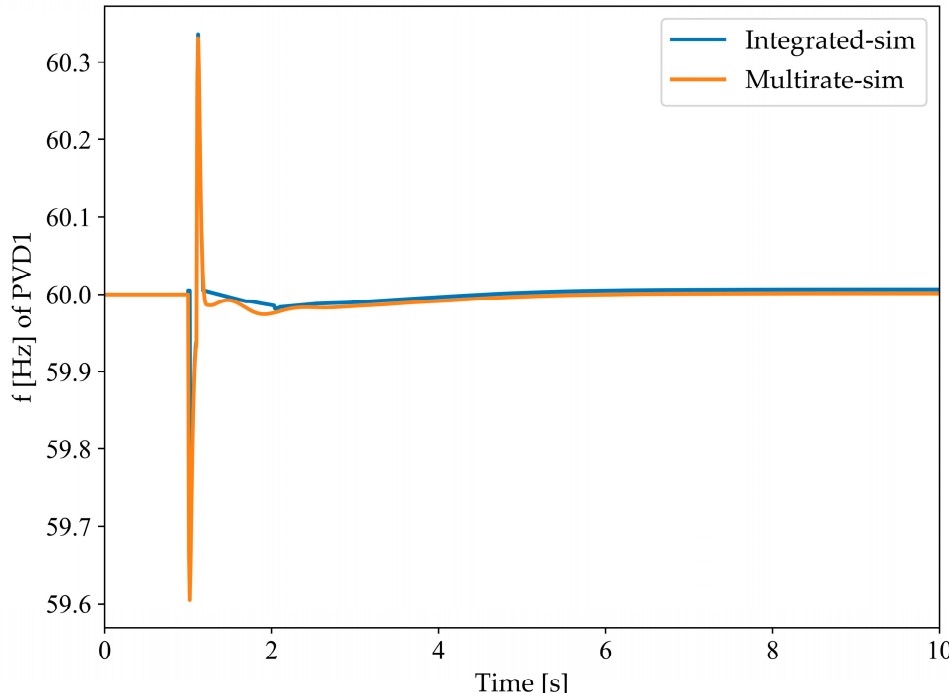

**Figure 7.** The frequency of PVD1 in the integrated and multi-rate simulations.

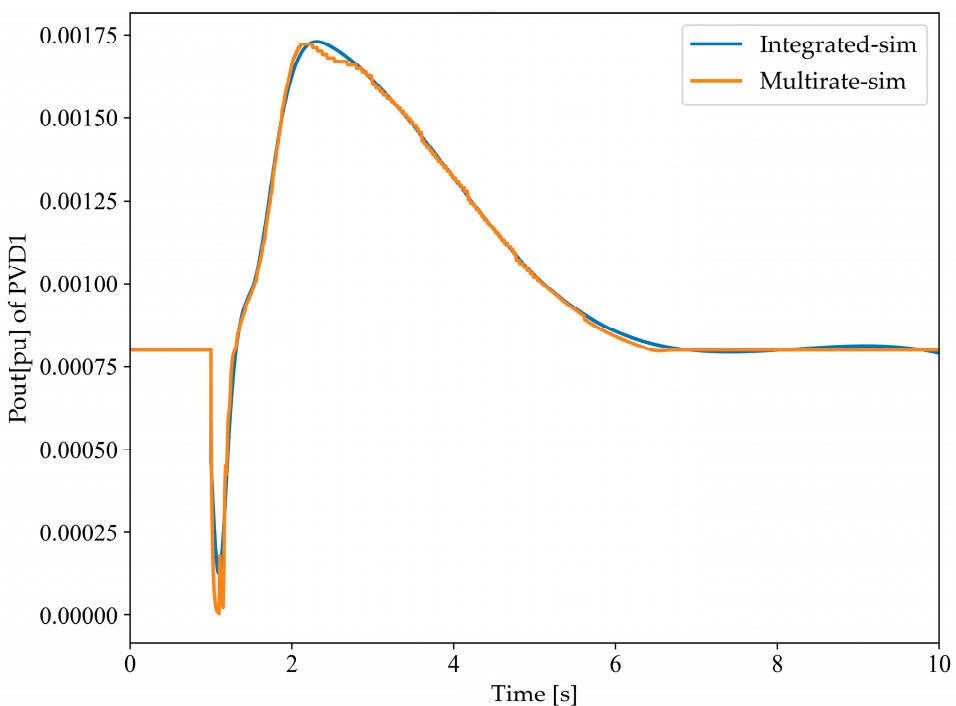

**Figure 8.** The output power of PVD1 in the integrated and multi-rate simulations.

**Table 1.** Simulation times of different scenarios in IEEE 14 + 33-bus system.

| Scenarios | T/s |
| --- | --- |
| Large step size | 3.4051 |
| Small step size | 142.0161 |
| Multi-rate | 66.7128 |

From Figures 5–7, it is evident that the multi-rate method captures the response of dynamic devices with greater accuracy. For instance, in Figure 5, the fluctuation process of the rotational speed during the rebalancing phase is depicted more clearly. Similarly, in Figure 8, the active power output of the photovoltaic System shows more detailed variations at the peaks and troughs.

From Table 1, it is observed that the entire integrated system, when simulated using a uniform large timestep, exhibits high efficiency with a simulation time of 3.4051 s. However, employing a uniform small timestep, although increasing the accuracy of the simulation, results in a longer duration, totaling 142.0161 s. The multi-rate method employed in this study manages to amalgamate the advantages of both approaches. While ensuring stability, it effectively reduces the simulation time by 53.03%, down to 66.7128 s, which is consistent with the earlier theoretical analysis, indicating the successful validation of the interface algorithm.

Additionally, as evidenced by Figure 9, the relative errors between the integrated and multi-rate simulations are within a reasonable range. The primary inaccuracies of the multi-rate method stem from Equations (11) and (12), which, after interpolation and correction, show no significant discrepancy when interfacing with the integrated System data. Table 1 reveals that the efficiency of the multi-rate simulation is markedly improved in comparison to the uniform small timestep approach, while its accuracy has been corroborated earlier. Hence, it is evident that the multi-rate algorithm proposed in this study successfully achieves a judicious balance between simulation accuracy and efficiency.

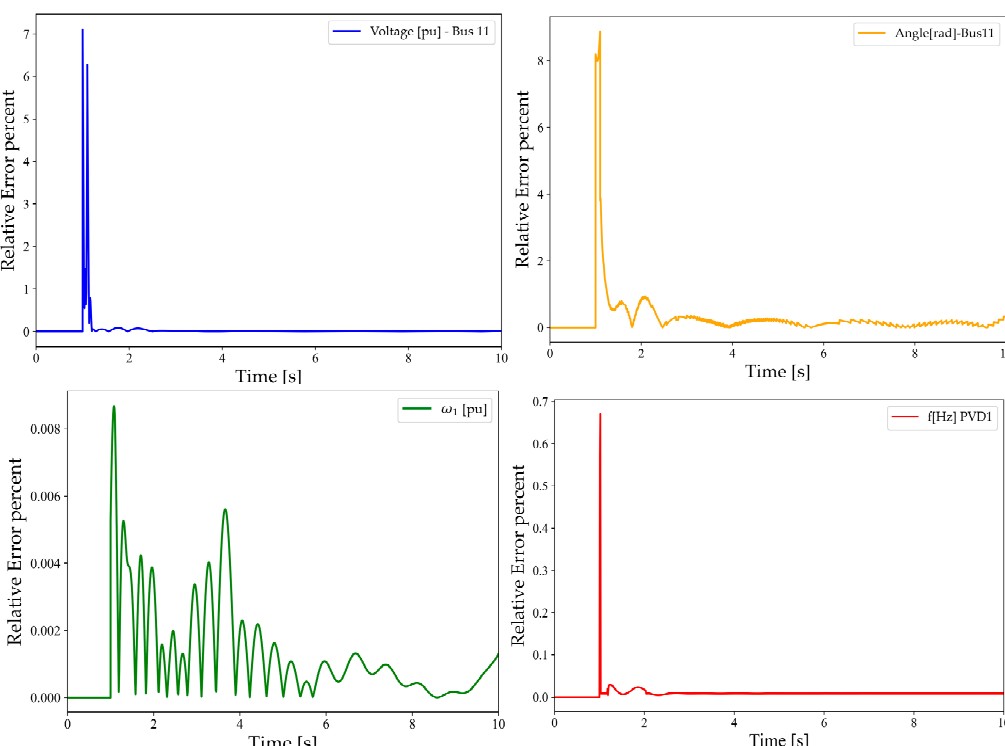

**Figure 9.** The relative error about variables in integrated and multi-rate simulations in IEEE 14-bus and 33-bus system.

### 3.2. Case Study of a Power System in the Guangxi Power Grid

3.2.1. Simulation Setup Description

To further illustrate the accuracy and applicability of the proposed multi-rate interface strategy, this case study utilizes a power System in the Guangxi Power Grid as a real-world example. After data conversion and topological reduction, this case can be equivalently represented as a 212-bus distribution System network, as shown in Figure 10. Bus 1 of the System is the substation, serving as the balance bus and is similarly considered as a synchronous generator, while all loads are modeled using a constant power model. Distributed energy storage devices are connected at Bus 10, 15, 106, 113, and 168 of the network. For the multi-rate method, the entire System is divided into two parts from Bus 98, which are jointly simulated using the aforementioned interface strategy. The static parameters of the network used in this study and the dynamic parameters of the energy storage devices are shown in GitHub [34]. During the simulation process, a three-phase short circuit fault lasting 0.1 s is introduced at Bus 59. Simulation step sizes are set at 10 ms and 1 ms, respectively, with a total simulation duration of 10 s. All results are presented in per-unit (pu) values.

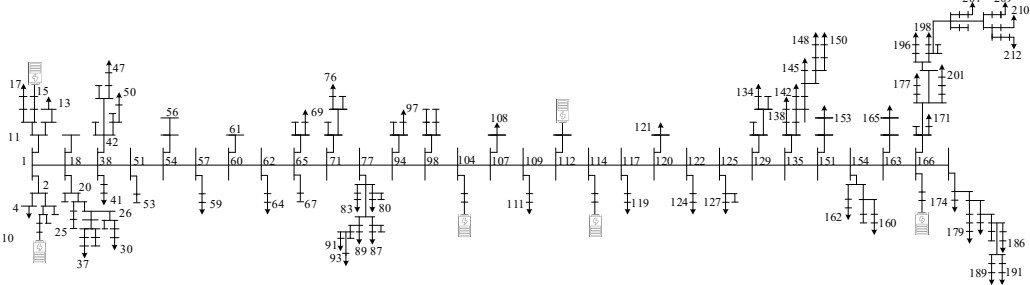

**Figure 10.** Topology of a power System in Guangxi Power Grid.

### 3.2.2. Simulation Results Analysis

As illustrated in Figures 11 and 12, the dynamic response of the distribution System network with integrated distributed energy storage devices exhibits greater fluctuations. In Figures 11 and 12, the active and reactive power outputs of the energy storage devices at Bus 174, simulated using both the uniform large step size method and the multi-rate method, are compared, yielding similarly favorable results.

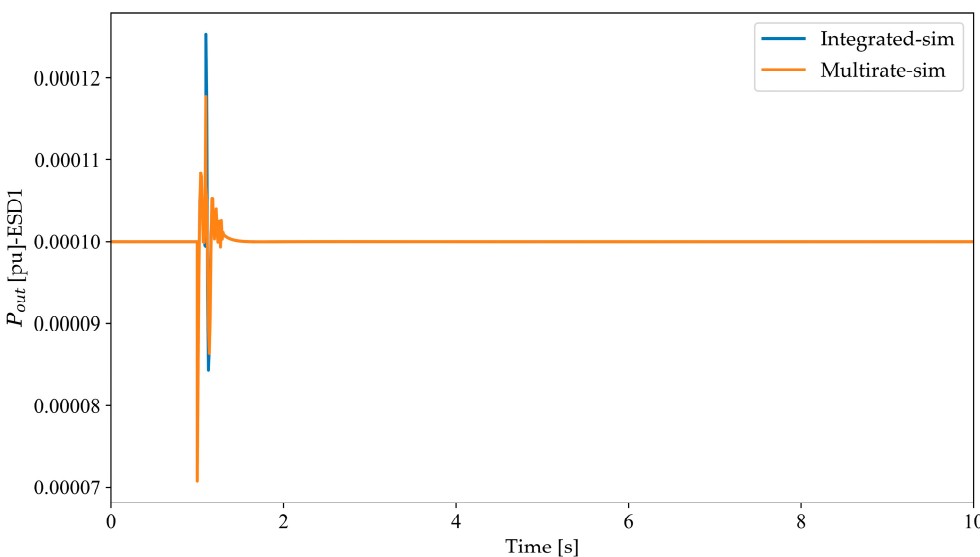

**Figure 11.** The output active power of the distributed energy storage device at Bus 174 in the integrated and multi-rate simulations.

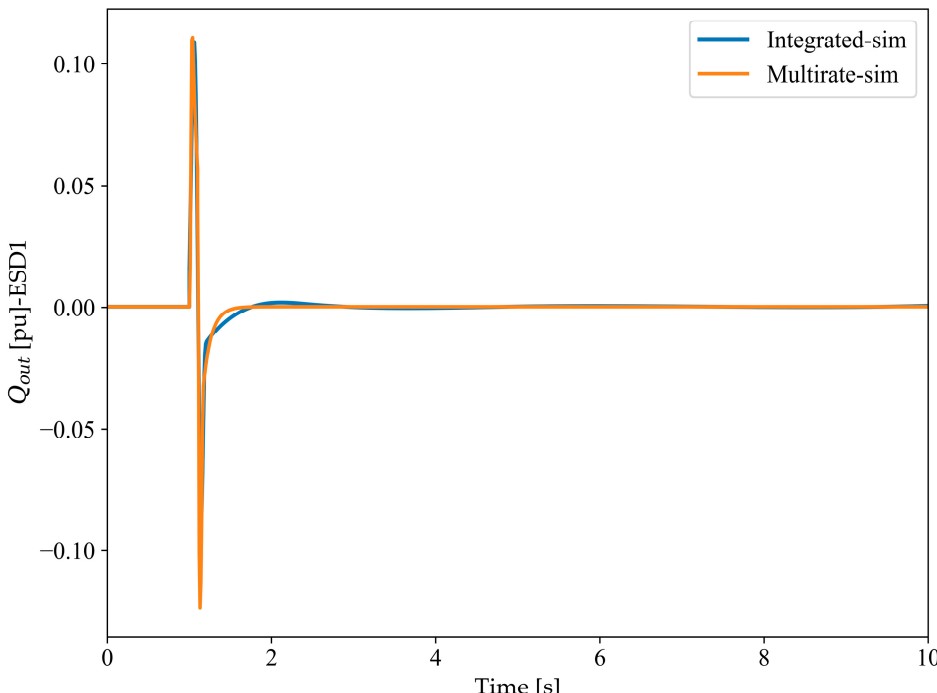

**Figure 12.** The output reactive power of the distributed energy storage device at Bus 174 in the integrated and multi-rate simulations.

Similarly, for this real-world System starting with 98 bus and partitioning, the order of the System matrix changes from $(2n)^2$ to $(0.85n)^2$ and $(1.15n)^2$. Therefore, based on

theoretical analysis, the efficiency of this System can be theoretically improved by 67.5%. It is noteworthy that the multi-rate method exhibits a more significant improvement in simulation efficiency for large-scale power systems. As indicated in Table 2, the uniform small timestep simulation takes a considerable amount of time, totaling 248.2992 s. However, while maintaining simulation accuracy, the multi-rate method substantially enhances the efficiency by 66.83%, ultimately reducing the simulation time to 82.3734 s. It is evident that the multi-rate interface strategy improves the efficiency of large-scale power systems more obviously.

**Table 2.** Simulation times of different scenarios of a power System in the Guangxi Power Grid.

| Scenarios | T/s |
|---|---|
| Large step size | 5.3975 |
| Small step size | 248.2992 |
| Multi-rate | 82.3734 |

### 4. Conclusions and Outlook

In this paper, a multi-rate simulation interface strategy is proposed, based on the modified time-domain fast simulation and multi-area data exchange method. As validated by the experiments, the following conclusions can be obtained.

(1) The paper introduces a modified TDS method. This method addresses inherent limitations encountered in traditional simulation software when applied to the multi-rate simulation. The proposed method eliminates the initialization process, allowing electric data to be directly exchanged within the System from the first exchange. This facilitates subsequent simulations to incrementally build upon initial results.

(2) The proposed multi-area data exchange method significantly simplifies exchange between different subsystems. Interpolation techniques are employed during the exchange process to correct numerical stability issues. Furthermore, the proposed method suggests a theoretical possibility by envisioning a general interface for different simulation software.

(3) The advantages of the proposed multi-rate interface strategy have been verified through a standard case study and a real-world power System in the Guangxi Power Grid in China. The standard case study shows that the simulation efficiency in the IEEE 14-bus and 33-bus System is improved by 53.03% compared with that of the integrated system, and is improved by 66.83% in the real-world system. Meanwhile, the simulation accuracy of the simulation is ensured.

Due to the current applicability limitations of multi-rate simulation interface and the idealized partitioning method, the focus for future research is outlined as follows:

(1) Abstracting the multi-rate simulation interface to meet the needs of the simulation framework, thereby enabling adaptation to other simulation software. This aims to reduce the need for custom modifications by users and enhance the scalability of the framework.

(2) Building upon the foundation of multi-rate simulation, incorporating dynamic partitioning strategies and parallel simulation techniques. This would better reflect the distribution of distributed energy devices in real power systems and further enhance the efficiency of the simulation process.

**Author Contributions:** Conceptualization, C.L. and Q.C.; methodology, Q.C. and Y.L.; software, Q.C.; validation, C.L., Q.C., H.B. and R.Y.; formal analysis, C.L., Q.C. and Y.L.; investigation, H.B. and R.Y.; resources, C.L., H.B., R.Y., T.L. and W.Y.; writing—original draft preparation, Q.C.; writing—review and editing, C.L. and Y.L.; supervision, C.L., Y.L., H.B. and R.Y.; project administration, H.B., R.Y., T.L. and W.Y. All authors have read and agreed to the published version of the manuscript.

**Funding:** This research was supported by the National Natural Science Foundation of China under Project 52007133.

**Data Availability Statement:** All data used to support the findings of this study are included within the article.

**Conflicts of Interest:** The authors declare no conflicts of interest.

## Appendix A

**Table A1.** The static active power (PL0), static reactive power (QL0) of the loads in IEEE 14 and 33-bus integrated system.

| Bus | $P_{L0}$ (pu) | $Q_{L0}$ (pu) |
|---|---|---|
| 2 | 0.217 | 0.127 |
| 3 | 0.5 | 0.25 |
| 4 | 0.478 | 0.1 |
| 5 | 0.076 | 0.016 |
| 6 | 0.15 | 0.075 |
| 9 | 0.295 | 0.166 |
| 10 | 0.09 | 0.058 |
| 11 | 0.035 | 0.018 |
| 12 | 0.061 | 0.016 |
| 13 | 0.135 | 0.058 |
| 14 | 0.2 | 0.07 |
| 16 | 0.001 | 0.0006 |
| 17 | 0.0009 | 0.0004 |
| 18 | 0.0012 | 0.0008 |
| 19 | 0.0006 | 0.0003 |
| 20 | 0.0006 | 0.0002 |
| 21 | 0.002 | 0.001 |
| 22 | 0.002 | 0.001 |
| 23 | 0.0006 | 0.0002 |
| 24 | 0.0006 | 0.0002 |
| 25 | 0.00045 | 0.0003 |
| 26 | 0.0006 | 0.00035 |
| 27 | 0.0006 | 0.00035 |
| 28 | 0.0012 | 0.0008 |
| 29 | 0.0006 | 0.0001 |
| 30 | 0.0006 | 0.0002 |
| 31 | 0.0006 | 0.0002 |
| 32 | 0.0009 | 0.0004 |
| 33 | 0.0009 | 0.0004 |
| 34 | 0.0009 | 0.0004 |
| 35 | 0.0009 | 0.0004 |
| 36 | 0.0009 | 0.0004 |
| 37 | 0.0009 | 0.0005 |
| 38 | 0.0042 | 0.002 |
| 39 | 0.0042 | 0.002 |
| 40 | 0.0006 | 0.00025 |
| 41 | 0.0006 | 0.00025 |
| 42 | 0.0006 | 0.0002 |
| 43 | 0.0012 | 0.0007 |
| 44 | 0.002 | 0.006 |
| 45 | 0.0015 | 0.0007 |
| 46 | 0.0021 | 0.001 |
| 47 | 0.0006 | 0.0004 |

**Table A2.** The static active power (PG0), static reactive power (QG0) of the generators in IEEE 14 + 33-bus system.

| Bus | $P_{G0}$ (pu) | $Q_{G0}$ (pu) |
|---|---|---|
| 2 | 0.4 | 0.15 |
| 3 | 0.4 | 0.15 |
| 6 | 0.3 | 0.1 |
| 8 | 0.35 | 0.1 |
| 27 | 0.008 | 0.006 |
| 33 | 0.002 | 0.008 |
| 37 | 0.004 | 0.004 |
| 40 | 0.008 | 0.004 |

**Table A3.** The partial dynamic parameters of the synchronous generators in IEEE 14 + 33-bus system.

| idx | D | M | ra | xl | xq | xd | xd1 | xd2 | xq1 | xq2 | Td10 | Td20 | Tq10 |
|---|---|---|---|---|---|---|---|---|---|---|---|---|---|
| 1 | 0 | 13 | 0 | 0.06 | 1.7 | 1.8 | 0.3 | 0.25 | 0.55 | 0.25 | 8 | 0.03 | 0.4 |
| 2 | 0 | 13 | 0 | 0.054 | 1.66 | 1.66 | 0.25 | 0.25 | 0.55 | 0.25 | 8 | 0.03 | 0.4 |
| 3 | 0 | 10 | 0 | 0.06 | 1.7 | 1.8 | 0.3 | 0.25 | 0.55 | 0.25 | 8 | 0.03 | 0.4 |
| 4 | 0 | 10 | 0 | 0.054 | 1.66 | 1.66 | 0.25 | 0.25 | 0.55 | 0.25 | 8 | 0.03 | 0.4 |
| 5 | 0 | 10 | 0 | 0.06 | 1.7 | 1.8 | 0.3 | 0.25 | 0.55 | 0.25 | 8 | 0.03 | 0.4 |

**Table A4.** Symbol description of dynamic parameters of the synchronous generator in IEEE 14 + 33-bus system.

| Symbol | Description |
|---|---|
| D | damping coefficient |
| M | machine start-up time |
| ra | armature resistance |
| xl | leakage reactance |
| xq | d-axis transient reactance |
| xd | d-axis synchronous reactance |
| xd1 | q-axis synchronous reactance |
| xd2 | d-axis sub-transient reactance |
| xq1 | q-axis transient reactance |
| xq2 | q-axis sub-transient reactance |
| Td10 | d-axis transient time constant |
| Td20 | d-axis sub-transient time constant |
| Tq10 | q-axis transient time constant |

**Table A5.** The partial dynamic parameters of the distributed PV in IEEE 14 + 33-bus system.

| idx | Bus | Sn | dqdv | fdbd | ddn | ialim | vt0 | vt1 | vt2 | vt3 | ft0 | ft1 | ft2 |
|---|---|---|---|---|---|---|---|---|---|---|---|---|---|
| 1 | 27 | 1 | 0 | 0.06 | −1 | −0.017 | 5 | 1.2 | 0.88 | 0.9 | 49.5 | 49.7 | 50.3 |
| 2 | 33 | 1 | 0 | 0.06 | −1 | −0.017 | 5 | 1.2 | 0.88 | 0.9 | 49.5 | 49.7 | 50.3 |
| 3 | 37 | 1 | 0 | 0.06 | −1 | −0.017 | 5 | 1.2 | 0.88 | 0.9 | 49.5 | 49.7 | 50.3 |
| 4 | 40 | 1 | 0 | 0.06 | −1 | −0.017 | 5 | 1.2 | 0.88 | 0.9 | 49.5 | 49.7 | 50.3 |

**Table A6.** Symbol description of dynamic parameters of the distributed PV in IEEE 14 + 33-bus system.

| Symbol | Description |
|---|---|
| dqdv | Q-V droop characteristics |
| fdbd | Frequency deviation deadband |
| ddn | Gain after f deadband |
| ialim | Apparent power limit |
| vt0 | Voltage tripping response curve point 0 |

**Table A6.** *Cont.*

| Symbol | Description |
|--------|-------------|
| vt1 | Voltage tripping response curve point 1 |
| vt2 | Voltage tripping response curve point 2 |
| vt3 | Voltage tripping response curve point 3 |
| ft0 | Frequency tripping response curve point 1 |
| ft1 | Frequency tripping response curve point 2 |
| ft2 | Frequency tripping response curve point 3 |

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
