# Peer review of "A Multi-Rate Simulation Strategy Based on the Modified Time-Domain Simulation Method and Multi-Area Data Exchange Method of Power Systems"

_electronics, doi:10.3390/electronics13050884_

Round 1
Reviewer 1 Report
Comments and Suggestions for Authors
In my opinion the paper is very interesting and we can find a lot of possible applications not only for what concern simulation of large plant.
In my opinion the paper must better specify the topic and the focus of the proposed solution. is it a fast simulation or a RT system?
Is it an hard or a soft real time systems?.
Finally is it suitable to compensate comunication delay for distributed remotely connected labs?
For what concern fast simulation on a single computational platform commercial softwares like Simulink, support the simulation of multiple subsystems with different variable and fixed step solvers also managing data exchange. Also Simulink through data store write and data store read can support this flexible approach quite easily
For RT processes if there are no comunication lags (as example all the devices in the same lab) you can use multiple or hybrid simulation platform (as example FPGA+CPU) .
So i suggest to better define the application and the intent of your methodology which in my opinion is better suited for Soft Real Time applications in which networks of remotely connected devices should tolerate also comunication delays (this is only an opinion/an impression you can be more clear and precise in the description if i am wrong)
Finally if your system is a soft real time or a fast simulation for diagnostic and monitoring (so some soft real time issue is still mantained) is it interesting to evaluate admissible jittering for your application considering that computational load is variable by definition in your method. Finally considering the wide variability of connected system to your plant you can consider some reference to bondgraph modelling techniques aiming to unify the modelling approach of multiphisical networks.
For what concern examples of bondagraph analogy i suggest this recent work in which the analogy between mechanical and electrical resonant systems is applied to resonant wireless system that are treated as corresponding mechanical vibrating ones.
Pugi, L., Reatti, A., Corti, F.
Application of modal analysis methods to the design of wireless power transfer systems
(2019) Meccanica, 54 (1-2), pp. 321-331.
DOI: 10.1007/s11012-018-00940-x
Reviewer 2 Report
Comments and Suggestions for Authors
The paper presents a multi-rate simulation interface strategy that boosts simulation efficiency while maintaining accuracy. Compared with other published materials the described method proposes a multi-area data exchange between different subsystems and an interpolation technique to solve the numerical stability issues.
The topic is relevant for the field and as the simulation results show it achieves an improvement in judicious balance between simulation accuracy and efficiency.
The work fulfills the stated objective, and the results have demonstrated that maintaining the accuracy of the simulation, the multi-rate method substantially improves the efficiency and reduces the simulation time.
The figures are clear, intuitive, and easy to interpret.
Tables A3 and A4 must be correctly aligned on the page.
The references are appropriate. However, some are quite old. More recent ones can be added. Use the same reference style for cited documents. The year of publication must appear on all titles (missing on [3], [33]).
The paper proposed a mathematical method applied to an energy power system. It does not fit entirely the Electronics journal scope.
